# Robust and Fully-Dynamic Coreset for Continuous-and-Bounded Learning (With Outliers) Problems

**Zixiu Wang**[1*]   **Yiwen Guo**[2]   **Hu Ding**[1†]
[1]School of Computer Science and Technology,
University of Science and Technology of China
[2]ByteDance AI Lab
wzx2014@mail.ustc.edu.cn, guoyiwen89@gmail.com, huding@ustc.edu.cn

## Abstract

In many machine learning tasks, a common approach for dealing with large-scale data is to build a small summary, *e.g.,* coreset, that can efficiently represent the original input. However, real-world datasets usually contain outliers and most existing coreset construction methods are not resilient against outliers (in particular, an outlier can be located arbitrarily in the space by an adversarial attacker). In this paper, we propose a novel robust coreset method for the *continuous-and-bounded learning* problems (with outliers) which includes a broad range of popular optimization objectives in machine learning, *e.g.,* logistic regression and $k$-means clustering. Moreover, our robust coreset can be efficiently maintained in fully-dynamic environment. To the best of our knowledge, this is the first robust and fully-dynamic coreset construction method for these optimization problems. Another highlight is that our coreset size can depend on the doubling dimension of the parameter space, rather than the VC dimension of the objective function which could be very large or even challenging to compute. Finally, we conduct the experiments on real-world datasets to evaluate the effectiveness of our proposed robust coreset method.

## 1 Introduction

As the rapid increasing of data volume in this big data era, we often need to develop low-complexity (*e.g.,* linear or even sublinear) algorithms for machine learning tasks. Moreover, our dataset is often maintained in a dynamic environment so that we have to consider the issues like data insertion and deletion. For example, as mentioned in the recent article [20], Ginart *et al.* discussed the scenario that some sensitive training data have to be deleted due to the reason of privacy preserving. Obviously, it is prohibitive to re-train our model when the training data is changed dynamically, if the data size is extremely large. To remedy these issues, a natural way is to construct a small-sized summary of the training data so that we can run existing algorithms on the summary rather than the whole data. **Coreset** [18], which was originally studied in the community of computational geometry [1], has become a widely used data summary for many large-scale machine learning problems [7, 27, 34, 38, 36, 25]. As a succinct data compression technique, coreset also enjoys several nice properties. For instance, coreset is usually composable and thus can be applied in the environment like distributed computing [28]. Also, it is usually able to obtain small coresets for streaming algorithms [23, 15] and fully-dynamic algorithms with data insertion and deletion [10, 24].

---

[*]Part of this work was done when Zixiu Wang was an intern under the supervision of Yiwen Guo at ByteDance.

[†]Corresponding author.

35th Conference on Neural Information Processing Systems (NeurIPS 2021).

However, the existing coreset construction methods are still far from being satisfactory in practice. A major bottleneck is that most of them are sensitive to outliers. We are aware that real-world dataset is usually noisy and may contain outliers; note that the outliers can be located arbitrarily in the space and even a single outlier can significantly destroy the final machine learning result. A typical example is poisoning attack, where an adversarial attacker may inject several specially crafted samples into the training data which can make the decision boundary severely deviate and cause unexpected misclassification [5]. In the past decades, a number of algorithms have been proposed for solving optimization with outliers problems, like clustering [11, 14, 12, 21, 42], regression [39, 37, 17], and PCA [8].

To see why the existing coreset methods are sensitive to outliers, we can take the popular sampling based coreset framework [19] as an example. The framework needs to compute a "sensitivity" for each data item, which measures the importance degree of the data item to the whole data set; however, it tends to assign high sensitivities to the points who are far from the majority of the data, that is, an outlier is likely to have a high sensitivity and thus has a high chance to be selected to the coreset. Obviously, the coreset obtained by this way is not pleasant since we expect to contain more inliers rather than outliers in the coreset. It is also more challenging to further construct a fully-dynamic robust coreset. The existing robust coreset construction methods [19, 26] often rely on simple uniform sampling and are efficient only when the number of outliers is a constant factor of the input size (we will discuss this issue in Section 3.1). Note that other outlier-resistant data summary methods like [21, 13] usually yield large approximation factors and are not easy to be maintained in a fully dynamic scenario, to our knowledge.

## 1.1 Our Contributions

In this paper, we propose a **unified fully-dynamic robust coreset framework** for a class of optimization problems which is termed *continuous-and-bounded (CnB) learning*. This type of learning problems covers a broad range of optimization objectives in machine learning [41, Chapter 12.2.2]. Roughly speaking, "CnB learning" requires that the optimization objective is a continuous function (*e.g.,* smooth or Lipschitz), and meanwhile the solution is restricted within a bounded region. We emphasize that this "bounded" assumption is quite natural in real machine learning scenarios. To shed some light, we can consider running an iterative algorithm (*e.g.,* the popular gradient descent or expectation maximization) for optimizing some objective; the solution is always restricted within a local region except for the first few rounds. Moreover, it is also reasonable to bound the solution range in a dynamic environment because one single update (insertion or deletion) is not likely to dramatically change the solution.

Our coreset construction is a novel **hybrid framework**. First, we suppose that there exists an ordinary coreset construction method $\mathcal{A}$ for the given CnB optimization objective (without considering outliers). Our key idea is to classify the input data into two parts: the "suspected" inliers and the "suspected" outliers, where the ratio of the sizes of these two parts is a carefully designed parameter $\lambda$. For the "suspected" inliers, we run the method $\mathcal{A}$ (as a black box); for the "suspected" outliers, we directly take a small sample uniformly at random; finally, we prove that these two parts together yield a robust coreset. Our framework can be also efficiently implemented under the merge-and-reduce framework for dynamic setting (though the original merge-and-reduce framework is not designed for the case with outliers) [4, 23]. A cute feature of our framework is that we can easily tune the parameter $\lambda$ for updating our coreset dynamically, if the fraction of outliers is changed in the dynamic environment.

The other contribution of this paper is that we propose two different coreset construction methods for CnB optimization objectives (*i.e.,* the aforementioned black box $\mathcal{A}$). The first method is based on the importance sampling framework [19], and the second one is based on a space partition idea. Our coreset sizes depend on the doubling dimension of the solution space rather than the VC (shattering) dimension. This property is particularly useful if the VC dimension is too high or not easy to compute, or considering the scenarios like sparse optimization (the domain of the solution vector has a low doubling dimension). To our knowledge, the only existing coreset construction methods that depend on doubling dimension are from Huang *et al.* [26] and Cohen-Addad *et al.* [16], but their results are only for clustering problems. Our methods can be applied for a broad range of widely studied optimization objectives, such as logistic regression [38], Bregman clustering [2], and truth discovery [31]. It is worth noting that although some coreset construction methods for them have

been proposed before (*e.g.,* [33, 38, 43, 17, 25]), they are all problem-dependent and we are the first, to the best of our knowledge, to study them from a unified "CnB" perspective.

## 2    Preliminaries

We introduce several important notations used throughout this paper. Suppose $\mathcal{P}$ is the parameter space. Let $X$ be the input data set that contains $n$ items in a metric space $\mathcal{X}$, and each $x \in X$ has a weight $w(x) \geq 0$. Further, we use $(X, z)$ to denote a given instance $X$ with $z$ outliers. We always use $|\cdot|$ and $[\![\cdot]\!]$ respectively to denote the number of data items and the total weight of a given data set. We consider the learning problem whose objective function is the weighted sum of the cost over $X$, *i.e.,*

$$f(\theta, X) := \sum_{x \in X} w(x) f(\theta, x), \tag{1}$$

where $f(\theta, x)$ is the non-negative cost contributed by $x$ with the parameter vector $\theta \in \mathcal{P}$. The goal is to find an appropriate $\theta$ so that the objective function $f(\theta, X)$ is minimized. Usually we assume each $x \in X$ has unit weight (*i.e.,* $w(x) = 1$), and it is straightforward to extend our method to weighted case. Given the pre-specified number $z \in \mathbb{Z}^+$ of outliers in $X$ (for weighted case, "$z$" refers to the total weight of outliers), we then define the "robust" objective function:

$$f_z(\theta, X) := \min_{O \subset X, [\![O]\!]=z} f(\theta, X \backslash O). \tag{2}$$

Actually, the above definition (2) comes from the popular "trimming" idea [39] that has been widely used for robust optimization problems.

Below we present the formal definition of continuous-and-bound learning problem. A function $g : \mathcal{P} \rightarrow \mathbb{R}$ is $\alpha$-*Lipschitz continuous* if for any $\theta_1, \theta_2 \in \mathcal{P}$, $|g(\theta_1) - g(\theta_2)| \leq \alpha \|\Delta\theta\|$, where $\Delta\theta = \theta_1 - \theta_2$ and $\|\cdot\|$ is some specified norm in $\mathcal{P}$.

**Definition 1** (Continuous-and-Bounded (CnB) Learning [41])**.** *Let $\alpha$, $\ell > 0$, and $\tilde{\theta} \in \mathcal{P}$. Denote by $\mathbb{B}(\tilde{\theta}, \ell)$ the ball centered at $\tilde{\theta}$ with radius $\ell$ in the parameter space $\mathcal{P}$. An objective (1) is called a CnB learning problem with the parameters $(\alpha, \ell, \tilde{\theta})$ if (i) the loss function $f(\cdot, x)$ is $\alpha$-Lipschitz continuous for any $x \in X$, and (ii) $\theta$ is always restricted within $\mathbb{B}(\tilde{\theta}, \ell)$.*

**Remark 1.** *We can also consider other variants for CnB learning with replacing the "$\alpha$-Lipschitz continuous" assumption. For example, a differentiable function $g$ is "$\alpha$-Lipschitz continuous gradient" if its gradient $\nabla g$ is $\alpha$-Lipschitz continuous (it is also called "$\alpha$-smooth"). Similarly, a twice-differentiable function $g$ is "$\alpha$-Lipschitz continuous Hessian" if its Hessian matrix $\nabla^2 g$ is $\alpha$-Lipschitz continuous. In this paper, we mainly focus the problems under the "$\alpha$-Lipschitz continuous" assumption, and our analysis can be also applied to these two variants via slight modifications.*

We define the coreset for CnB learning problems below.

**Definition 2** ($\varepsilon$-coreset)**.** *Let $\varepsilon > 0$. Given a dataset $X \subset \mathcal{X}$ and the objective function $f(\theta, X)$, we say that a weighted set $C \subset \mathcal{X}$ is an $\varepsilon$-coreset of $X$ if for any $\theta \in \mathbb{B}(\tilde{\theta}, \ell)$, we have*

$$|f(\theta, C) - f(\theta, X)| \leq \varepsilon f(\theta, X). \tag{3}$$

If $C$ is an $\varepsilon$-coreset of $X$, we can run an existing optimization algorithm on $C$ so as to obtain an approximate solution. Obviously, we expect that the size of $C$ to be as small as possible. Following Definition 2, we also define the corresponding *robust coreset* (the similar definition was also introduced in [19, 26] before).

**Definition 3** (robust coreset)**.** *Let $\varepsilon > 0$, and $\beta \in [0, 1)$. Given the dataset $X \subset \mathcal{X}$ and the objective function $f(\theta, x)$, we say that a weighted dataset $C \subset \mathcal{X}$ is a $(\beta, \varepsilon)$-robust coreset of $X$ if for any $\theta \in \mathbb{B}(\tilde{\theta}, \ell)$, we have*

$$(1 - \varepsilon) f_{(1+\beta)z}(\theta, X) \leq f_z(\theta, C) \leq (1 + \varepsilon) f_{(1-\beta)z}(\theta, X). \tag{4}$$

Roughly speaking, if we obtain an approximate solution $\theta' \in \mathcal{P}$ on $C$, its quality can be preserved on the original input data $X$. The parameter $\beta$ indicates the error on the number of outliers if using

$\theta'$ as our solution on $X$. If we set $\beta = 0$, that means we allow no error on the number of outliers. In our full paper [44], we present our detailed discussion on the quality loss (in terms of the objective value and the number of outliers) of this transformation from $C$ to $X$.

**The rest of this paper is organized as follows.** In Section 3, we introduce our robust coreset framework and show how to implement it in a fully-dynamic environment. In Section 4, we propose two different ordinary coreset (without outliers) construction methods for CnB learning problems, which can be used as the black box in our robust coreset framework of Section 3. Finally, in Section 5 we illustrate the application of our coreset method in practice. **Due to the space limit**, some proofs and the more detailed experimental results are placed in our full paper [44].

# 3 Our Robust Coreset Framework

We first consider the simple uniform sampling as the robust coreset in Section 3.1 (in this part, we consider the general learning problems without the CnB assumption). To improve the result, we further introduce our major contribution, the hybrid framework for robust coreset construction and its fully-dynamic realization in Section 3.2 and 3.3, respectively.

## 3.1 Uniform Sampling for General Case

As mentioned before, the existing robust coreset construction methods [19, 26] are based on uniform sampling. Note that their methods are only for the clustering problems (*e.g.,* $k$-means/median clustering). Thus a natural question is that whether the uniform sampling idea also works for the general learning problems in the form of (1). Below we answer this question in the affirmative. To illustrate our idea, we need the following definition for range space.

**Definition 4** ($f$-induced range space). *Suppose $\mathcal{X}$ is an arbitrary metric space. Given the cost function $f(\theta, x)$ as (1) over $\mathcal{X}$, we let*

$$\mathfrak{R} = \Big\{ \{x \in \mathcal{X} : f(\theta, x) \leq r\} \mid \forall r \geq 0, \forall \theta \in \mathcal{P} \Big\}, \tag{5}$$

*then $(\mathcal{X}, \mathfrak{R})$ is called the $f$-induced range space. Each $R \in \mathfrak{R}$ is called a range of $\mathcal{X}$.*

The following "$\delta$-sample" concept comes from the theory of VC dimension [32]. Given a range space $(\mathcal{X}, \mathfrak{R})$, let $C$ and $X$ be two finite subsets of $\mathcal{X}$. Suppose $\delta \in (0, 1)$. We say $C$ is a $\delta$-*sample* of $X$ if $C \subseteq X$ and

$$\left| \frac{|X \cap R|}{|X|} - \frac{|C \cap R|}{|C|} \right| \leq \delta \text{ for any } R \in \mathfrak{R}. \tag{6}$$

Denote by vcdim the VC dimension of the range space of Definition 4, then we can achieve a $\delta$-sample with probability $1 - \eta$ by uniformly sampling $O(\frac{1}{\delta^2}(\text{vcdim} + \log \frac{1}{\eta}))$ points from $X$ [32]. The value of vcdim depends on the function "$f$". For example, if "$f$" is the loss function of logistic regression in $\mathbb{R}^d$, then vcdim can be as large as $\Theta(d)$ [38]. The following theorem shows that a $\delta$-sample can serve as a robust coreset if $z$ is a constant factor of $n$. Note that in the following theorem, the objective $f$ can be any function without following Definition 1.

**Theorem 1.** *Let $(X, z)$ be an instance of the robust learning problem (2). If $C$ is a $\delta$-sample of $X$ in the $f$-induced range space. We assign $w(c) = \frac{n}{|C|}$ for each $c \in C$. Then we have*

$$f_{z+\delta n}(\theta, X) \leq f_z(\theta, C) \leq f_{z-\delta n}(\theta, X) \tag{7}$$

*for any $\theta \in \mathcal{P}$ and any $\delta \in (0, z/n]$. In particular, if $\delta = \beta z/n$, $C$ is a $(\beta, 0)$-robust coreset of $X$ and the size of $C$ is $O(\frac{1}{\beta^2}(\frac{n}{z})^2(\text{vcdim} + \log \frac{1}{\eta}))$.*

*Proof.* Suppose the size of $C$ is $m$ and let $N = nm$. To prove Theorem 1, we imagine to generate two new sets as follows. For each point $c \in C$, we generate $n$ copies; consequently we obtain a new set $C'$ that actually is the union of $n$ copies of $C$. Similarly, we generate a new set $X'$ that is the union of $m$ copies of $X$. Obviously, $|C'| = |X'| = N$. Below, we fix an arbitrary $\theta \in \mathcal{P}$ and show that (7) is true.

We order the points of $X'$ based on their objective values; namely, $X' = \{x_i' \mid 1 \leq i \leq N\}$ and $f(\theta, x_1') \leq f(\theta, x_2') \leq \cdots \leq f(\theta, x_N')$. Similarly, we have $C' = \{c_i' \mid 1 \leq i \leq N\}$ and

$f(\theta, c'_1) \leq f(\theta, c'_2) \leq \cdots \leq f(\theta, c'_N)$. Then we claim that for any $1 \leq i \leq (1-\delta)N$, the following inequality holds:

$$f(\theta, c'_{i+\delta N}) \geq f(\theta, x'_i). \tag{8}$$

Otherwise, there exists some $i_0$ that $f(\theta, c'_{i_0+\delta N}) < f(\theta, x'_{i_0})$. Consider the range $R_0 = \{x \in \mathcal{X} \mid f(\theta, x) \leq f(\theta, c'_{i_0+\delta N})\}$. Then we have

$$\frac{|C \cap R_0|}{|C|} = \frac{|C' \cap R_0|/n}{|C|} \quad \geq \quad \frac{(i_0 + \delta N)/n}{m} = \frac{i_0}{N} + \delta; \tag{9}$$

$$\frac{|X \cap R_0|}{|X|} = \frac{|X' \cap R_0|/m}{|X|} \quad < \quad \frac{i_0/m}{n} = \frac{i_0}{N}. \tag{10}$$

That is, $\left| \frac{|C \cap R_0|}{|C|} - \frac{|X \cap R_0|}{|X|} \right| > \delta$ which is in contradiction with the fact that $C$ is a $\delta$-sample of $X$. Thus (8) is true. As a consequence, we have

$$f_z(\theta, C) \quad = \quad \frac{n}{m} \sum_{i=1}^{m-\frac{m}{n}z} f(\theta, c_i) = \frac{1}{m} \sum_{i=1}^{N-mz} f(\theta, c'_i) \geq \frac{1}{m} \sum_{i=1+\delta N}^{N-mz} f(\theta, c'_i) \tag{11}$$

$$\underbrace{\geq}_{\text{by (8)}} \quad \frac{1}{m} \sum_{i=1}^{(1-\delta)N-mz} f(\theta, x'_i) = \sum_{i=1}^{(1-\delta)n-z} f(\theta, x_i) = f_{z+\delta n}(\theta, X). \tag{12}$$

So the left-hand side of (7) is true, and the right-hand side can be proved by using the similar manner. □

**Remark 2.** *Our proof is partly inspired by the ideas of [37, 35] for analyzing uniform sampling. Though the uniform sampling is simple and easy to implement, it has two major drawbacks. First, it always involves an error "$\delta$" on the number of outliers (otherwise, if letting $\delta = 0$, the sample should be the whole $X$). Also, the result is interesting only when $z$ is a constant factor of $n$. For example, if $z = \sqrt{n}$, the obtained sample size can be as large as $n$. Our hybrid robust framework proposed in Section 3.2 can resolve these two issues for CnB learning problems.*

### 3.2 The Hybrid Framework for $(\beta, \varepsilon)$-Robust Coreset

Our idea for building the robust coreset comes from the following intuition. In an ideal scenario, if we know who are the inliers and who are the outliers, we can simply construct the coresets for them separately. In reality, though we cannot obtain such a clear classification, the CnB property (Definition 1) can guide us to obtain a "coarse" classification. Furthermore, together with some novel insights in geometry, we prove that such a hybrid framework can yield a $(\beta, \varepsilon)$-robust coreset.

Suppose $\varepsilon \in (0, 1)$ and the objective $f$ is continuous-and-bounded as Definition 1. Specifically, the parameter vector $\theta$ is always restricted within the ball $\mathbb{B}(\tilde{\theta}, \ell)$. First, we classify $X$ into two parts according to the value of $f(\tilde{\theta}, x)$. Let $\varepsilon_0 := \min \left\{ \frac{\varepsilon}{16}, \frac{\varepsilon \cdot \inf_{\theta \in \mathbb{B}(\tilde{\theta}, \ell)} f_z(\theta, X)}{16(n-z)\alpha\ell} \right\}$ and $\tilde{z} := (1 + 1/\varepsilon_0) z$; also assume $x_{\tilde{z}} \in X$ is the point who has the $\tilde{z}$-th largest cost $f(\tilde{\theta}, x)$ among $X$. We let $\tau = f(\tilde{\theta}, x_{\tilde{z}})$, and thus we obtain the set

$$\{x \in X \mid f(\tilde{\theta}, x) \geq \tau\}. \tag{13}$$

that has size $\tilde{z}$. We call these $\tilde{z}$ points as the "suspected outliers" (denoted as $X_{\text{so}}$) and the remaining $n - \tilde{z}$ points as the "suspected inliers" (denoted as $X_{\text{si}}$). If we fix $\theta = \tilde{\theta}$, the set of the "suspected outliers" contains at least $\frac{1}{\varepsilon_0} z$ real inliers (since $\tilde{z} = (1 + 1/\varepsilon_0) z$). This immediately implies the following inequality:

$$\tau z \leq \varepsilon_0 f_z(\tilde{\theta}, X). \tag{14}$$

Suppose we have an ordinary coreset construction method $\mathcal{A}$ as the black box (we will discuss it in Section 4). **Our robust coreset construction** is as follows:

*We build an $\varepsilon_1$-coreset ($\varepsilon_1 = \varepsilon/4$) for $X_{\text{si}}$ by $\mathcal{A}$ and take a $\delta$-sample for $X_{\text{so}}$ with setting $\delta = \frac{\beta\varepsilon_0}{1+\varepsilon_0}$. We denote these two sets as $C_{\text{si}}$ and $C_{\text{so}}$ respectively. If we set $\beta = 0$ (i.e., require no error on the number of outliers), we just directly take all the points of $X_{\text{so}}$ as $C_{\text{so}}$. Finally, we return $C = C_{\text{si}} \cup C_{\text{so}}$ as the robust coreset.*

**Theorem 2.** *Given a CnB learning instance $(X, z)$, the above coreset construction method returns a $(\beta, \varepsilon)$-robust coreset (as Defintion 3) of size*

$$|C_{\mathtt{si}}| + \min \left\{ O\left( \frac{1}{\beta^2 \varepsilon^2} (\mathtt{vcdim} + \log \frac{1}{\eta}) \right), O\left( \frac{z}{\varepsilon_0} \right) \right\} \tag{15}$$

*with probability at least $1 - \eta$. In particular, when $\beta = 0$, our coreset has no error on the number of outliers and its size is $|C_{\mathtt{si}}| + O\left( \frac{z}{\varepsilon_0} \right)$.*

*Proof.* **(sketch)** It is easy to obtain the coreset size. So we only focus on proving the quality guarantee below.

Let $\theta$ be any parameter vector in the ball $\mathbb{B}(\tilde{\theta}, \ell)$. Similar with the aforementioned classification $X_{\mathtt{si}} \cup X_{\mathtt{so}}$, $\theta$ also yields a classification on $X$. Suppose $\tau_\theta$ is the $z$-th largest value of $\{ f(\theta, x) \mid x \in X \}$. Then we use $X_{\mathtt{ri}}$ to denote the set of $n - z$ "real" inliers with respect to $\theta$, *i.e.,* $\{ x \in X \mid f(\theta, x) < \tau_\theta \}$; we also use $X_{\mathtt{ro}}$ to denote the set of $z$ "real" outliers with respect to $\theta$, *i.e.,* $\{ x \in X \mid f(\theta, x) \geq \tau_\theta \}$. Overall, the input set $X$ is partitioned into 4 parts:

$$X_{\mathrm{I}} = X_{\mathtt{si}} \cap X_{\mathtt{ri}}, X_{\mathrm{II}} = X_{\mathtt{so}} \cap X_{\mathtt{ri}}, X_{\mathrm{III}} = X_{\mathtt{so}} \cap X_{\mathtt{ro}}, \text{ and } X_{\mathrm{IV}} = X_{\mathtt{si}} \cap X_{\mathtt{ro}}. \tag{16}$$

Similarly, $\theta$ also yields a classification on $C$ to be $C_{\mathtt{ri}}$ (the set of "real" inliers of $C$) and $C_{\mathtt{ro}}$ (the set of "real" outliers of $C$). Therefore we have

$$C_{\mathrm{I}} = C_{\mathtt{si}} \cap C_{\mathtt{ri}}, C_{\mathrm{II}} = C_{\mathtt{so}} \cap C_{\mathtt{ri}}, C_{\mathrm{III}} = C_{\mathtt{so}} \cap C_{\mathtt{ro}}, \text{ and } C_{\mathrm{IV}} = C_{\mathtt{si}} \cap C_{\mathtt{ro}}. \tag{17}$$

Our goal is to show that $f_z(\theta, C)$ is a qualified approximation of $f_z(\theta, X)$ (as Defintion 3). Note that $f_z(\theta, C) = f(\theta, C_{\mathrm{I}} \cup C_{\mathrm{II}}) = f(\theta, C_{\mathrm{I}}) + f(\theta, C_{\mathrm{II}})$. Hence we can bound $f(\theta, C_{\mathrm{I}})$ and $f(\theta, C_{\mathrm{II}})$ separately. We consider their upper bounds first, and the lower bounds can be derived by using the similar manner.

The upper bound of $f(\theta, C_{\mathrm{I}})$ directly comes from the definition of $\varepsilon$-coreset, *i.e.,* $f(\theta, C_{\mathrm{I}}) \leq f(\theta, C_{\mathrm{I}} \cup C_{\mathrm{IV}}) \leq (1 + \varepsilon_1) f(\theta, X_{\mathrm{I}} \cup X_{\mathrm{IV}})$ since $C_{\mathrm{I}} \cup C_{\mathrm{IV}}$ is an $\varepsilon_1$-coreset of $X_{\mathrm{I}} \cup X_{\mathrm{IV}}$.

It is more complicated to derive the upper bound of $f(\theta, C_{\mathrm{II}})$. We consider two cases. **(1)** If $C_{\mathrm{IV}} = \emptyset$, then we know that all the suspected inliers of $C$ are all real inliers (and meanwhile, all the real outliers of $C$ are suspected outliers); consequently, we have

$$f(\theta, C_{\mathrm{II}}) \leq f_z(\theta, C_{\mathrm{II}} \cup C_{\mathrm{III}}) \leq f_{(1-\beta)z}(\theta, X_{\mathrm{II}} \cup X_{\mathrm{III}}) \tag{18}$$

from Theorem 1. **(2)** If $C_{\mathrm{IV}} \neq \emptyset$, by using the triangle inequality and the $\alpha$-Lipschitz assumption, we have $f(\theta, C_{\mathrm{II}}) \leq f(\theta, X_{\mathrm{II}}) + 2z(\tau + \alpha\ell))$. We merge these two cases and overall obtain the following upper bound:

$$f_z(\theta, C) \leq (1 + \varepsilon_1) f_{(1-\beta)z}(\theta, X) + 4z\tau + 4z\alpha\ell. \tag{19}$$

Moreover, from (14) and the $\alpha$-Lipschitz assumption, we have $\tau z \leq \varepsilon_0 f_z(\theta, X) + \varepsilon_0 (n - z)\alpha\ell$. Then the above (19) implies

$$f_z(\theta, C) \leq (1 + \varepsilon) f_{(1-\beta)z}(\theta, X). \tag{20}$$

Similarly, we can obtain the lower bound

$$f_z(\theta, C) \geq (1 - \varepsilon) f_{(1+\beta)z}(\theta, X). \tag{21}$$

Therefore $C$ is a $(\beta, \varepsilon)$-robust coreset of $X$. $\qquad\square$

### 3.3 The Fully-Dynamic Implementation

In this section, we show that our robust coreset of Section 3.2 can be efficiently implemented in a fully-dynamic environment, even if the number of outliers $z$ is dynamically changed.

The standard $\varepsilon$-coreset usually has two important properties. If $C_1$ and $C_2$ are respectively the $\varepsilon$-coresets of two disjoint sets $X_1$ and $X_2$, their union $C_1 \cup C_2$ should be an $\varepsilon$-coreset of $X_1 \cup X_2$. Also, if $C_1$ is an $\varepsilon_1$-coreset of $C_2$ and $C_2$ is an $\varepsilon_2$-coreset of $C_3$, $C_1$ should be an $(\varepsilon_1 + \varepsilon_2 + \varepsilon_1\varepsilon_2)$-coreset of $C_3$. Based on these two properties, one can build a coreset for incremental data stream by using

the "merge-and-reduce" technique [4, 23]. Very recently, Henzinger and Kale [24] extended it to the more general fully-dynamic setting, where data items can be deleted and updated as well.

Roughly speaking, the merge-and-reduce technique uses a sequence of "buckets" to maintain the coreset for the input streaming data, and the buckets are merged by a bottom-up manner. However, it is challenging to directly adapt this strategy to the case with outliers, because we cannot determine the number of outliers in each bucket. A cute aspect of our hybrid robust coreset framework is that we can easily resolve this obstacle by using an $O(n)$ size auxiliary table $\mathscr{L}$ together with the merge-and-reduce technique (note that even for the case without outliers, maintaining a fully-dynamic coreset already needs $\Omega(n)$ space [24]). We briefly introduce our idea below.

Recall that we partition the input data $X$ into two parts: the $n - \tilde{z}$ "suspected inliers" and the $\tilde{z}$ "suspected outliers", where $\tilde{z} = (1 + 1/\varepsilon_0)z$. We follow the same notations used in Section 3.2. For the first part, we just apply the vanilla merge-and-reduce technique to obtain a fully-dynamic coreset $C_{\mathtt{si}}$; for the other part, we can just take a $\delta$-sample or take the whole set (if we require $\beta$ to be 0), and denote it as $C_{\mathtt{so}}$. Moreover, we maintain a table $\mathscr{L}$ to record the key values $x.\mathtt{value} = f(\tilde{\theta}, x)$ and its position $x.\mathtt{position}$ in the merge-and-reduce tree, for each $x \in X$; they are sorted by the $x.\mathtt{values}$ in the table. To deal with the dynamic updates (*e.g.,* deletion and insertion), we also maintain a critical pointer $p$ pointing to the

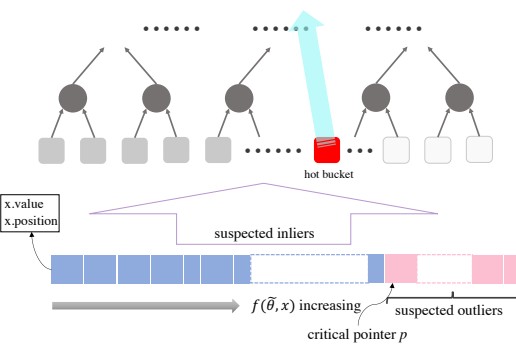

Figure 1: The illustration for our fully-dynamic robust coreset construction.

data item $x_{\tilde{z}}$ (recall $x_{\tilde{z}}$ has the $\tilde{z}$-th largest cost $f(\tilde{\theta}, x)$ among $X$ defined in Section 3.2).

When a new data item $x$ is coming or an existing data item $x$ is going to be deleted, we just need to compare it with $f(\tilde{\theta}, x_c)$ so as to decide to update $C_{\mathtt{si}}$ or $C_{\mathtt{so}}$ accordingly; after the update, we also need to update $x_{\tilde{z}}$ and the pointer $p$ in $\mathscr{L}$. If the number of outliers $z$ is changed, we just need to update $x_{\tilde{z}}$ and $p$ first, and then update $C_{\mathtt{si}}$ and $C_{\mathtt{so}}$ (for example, if $z$ is increased, we just need to delete some items from $C_{\mathtt{so}}$ and insert some items to $C_{\mathtt{si}}$). To realize these updating operations, we also set one bucket as the "hot bucket", which serves as a shuttle to execute all the data shifts. See Figure 1 for the illustration. Let $M(\varepsilon)$ be the size of the vanilla $\varepsilon$-coreset. In order to achieve an $\varepsilon$-coreset overall, we need to construct an $\frac{\varepsilon}{\log n}$-coreset with size $M(\varepsilon/\log n)$ in every reduce part [1]. We use $M$ to denote $M(\varepsilon/\log n)$ for short and assume that we can compute a coreset of $X$ in time $\mathtt{t}(|X|)$ [40], then we have the following result.

**Theorem 3.** *In our dynamic implementation, the time complexity for insertion and deletion is* $O(\mathtt{t}(M) \log n)$. *To update $z$ to $z \pm \Delta z$ with $\Delta z \geq 0$, the time complexity is* $O(\frac{\Delta z}{\varepsilon} \mathtt{t}(M) \log n)$, *where $\varepsilon$ is the error bound for the robust coreset in Definition 3.*

## 4 Coreset for Continuous-and-Bounded Learning Problems

As mentioned in Section 3.2, we need a black-box ordinary coreset (without considering outliers) construction method $\mathcal{A}$ in the hybrid robust coreset framework. In this section, we provide two different $\varepsilon$-coreset construction methods for the CnB learning problems.

### 4.1 Importance Sampling Based Coreset Construction

We follow the importance sampling based approach [30]. Suppose $X = \{x_1, \cdots, x_n\}$. For each data point $x_i$, it has a sensitivity $\sigma_i = \sup_\theta \frac{f(\theta, x)}{f(\theta, X)}$ that measures its importance to the whole input data $X$. Computing the sensitivity is often challenging but an upper bound of the sensitivity actually is already sufficient for the coreset construction. Assume $s_i$ is an upper bound of $\sigma_i$ and let $S = \sum_{i=1}^n s_i$. The coreset construction is as follows. We sample a subset $C$ from $X$, where each element of $C$ is sampled *i.i.d.* with probability $p_i = s_i/S$; we assign a weight $w_i = \frac{S}{s_i |C|}$ to each sampled data item $x_i$ of $C$. Finally, we return $C$ as the coreset.

**Theorem 4** ([7]). *Let* vcdim *be the VC dimension (or shattering dimension) of the range space induced from* $f(\theta, x)$. *If the size of* $C$ *is* $\Theta\left(\frac{S}{\varepsilon^2}\left(\text{vcdim} \cdot \log S + \log \frac{1}{\eta}\right)\right)$, *then* $C$ *is an* $\varepsilon$-*coreset with probability at least* $1 - \eta$.

Therefore the only remaining issue is how to compute the upper bounds $s_i$s. Recall that we assume our cost function is $\alpha$-Lipschitz (or $\alpha$-smooth, $\alpha$-Lipschitz continuous Hessian) in Definition 1. That is, we can bound the difference between $f(\theta, x_i)$ and $f(\tilde{\theta}, x_i)$, and such a bound can help us to compute $s_i$. In our full paper [44], we show that computing $s_i$ is equivalent to solving a **quadratic fractional programming**. This programming can be reduced to a semi-definite programming (SDP) [3], which can be solved in polynomial time up to any desired accuracy [22]. We denote the solving time of SDP by $\mathsf{T}(d)$, where $d$ is the dimension of the data point. So the total running time of the coreset construction is $O(n \cdot \mathsf{T}(d))$.

A drawback of Theorem 4 is that the coreset size depends on vcdim induced by $f(\theta, x)$. For some objectives, the value vcdim can be very large or difficult to obtain. Here, we prove that for a continuous-and-bounded cost function, the coreset size can be independent of vcdim; instead, it depends on the doubling dimension ddim [9] of the parameter space $\mathcal{P}$. Doubling dimension is a widely used measure to describe the growth rate of the data, which can also be viewed as a generalization of the Euclidean dimension. For example, the doubling dimension of a $d$-dimensional Euclidean space is $\Theta(d)$.

**Theorem 5.** *Given a CnB learning instance* $X$ *with the objective function* $f(\theta, X)$ *as described in Definition 1, let* ddim *be the doubling dimension of the parameter space. Then, if we run the importance sampling based coreset construction method with the sample size* $|C| = \Theta\left(\frac{S^2}{\varepsilon^2}\left(\text{ddim} \cdot \log \frac{1}{\varepsilon} + \log \frac{1}{\eta}\right)\right)$, $C$ *will be an* $\varepsilon$-*coreset with probability* $1 - \eta$. *The hidden constant of* $|C|$ *depends on the Lipschitz constant* $\alpha$ *and* $\inf_{\theta \in \mathbb{B}(\tilde{\theta}, \ell)} \frac{1}{n} f(\theta, X)^3$.

The major advantage of Theorem 5 over Theorem 4 is that we do not need to know the VC dimension induced by the cost function. On the other hand, the doubling dimension is often much easier to know (or estimate), *e.g.,* the doubling dimension of a given instance in $\mathbb{R}^d$ is just $\Theta(d)$, even the cost function can be very complicated. Another motivation of Theorem 5 is from sparse optimization. Let the parameter space be $\mathbb{R}^D$, and we restrict $\theta$ to be $k$-sparse (*i.e.*, at most $k$ non-zero entries with $k \ll D$). It is easy to see the domain of $\theta$ is a union of $\binom{D}{k}$ $k$-dimensional subspaces, and thus its doubling dimension is $O(k \log D)$ which is much smaller than $D$ (each ball of radius $r$ in the domain can be covered by $\binom{D}{k} \cdot 2^{O(k)} = 2^{O(k \log D)}$ balls of radius $r/2$).

The reader is also referred to [26] for a more detailed discussion on the relation between VC (shattering) dimension and doubling dimension.

## 4.2 Spatial Partition Based Coreset Construction

The reader may realize that the coreset size presented in Theorem 5 (and also Theorem 4) is **data-dependent**. That is, the coreset size depends on the value $S$, which can be different for different input instances. To achieve a **data-independent** coreset size, we introduce the following method based on spatial partition, which is partly inspired by the previous $k$-median/means clustering coreset construction idea of [15, 17, 25]. We generalize their method to the continuous-and-bounded learning problems and call it as *Generalized Spatial Partition (GSP)* method.

**GSP coreset construction.** We set $\varrho = \min_{x \in X} f(\tilde{\theta}, x)$ and $T = \frac{1}{|X|} f(\tilde{\theta}, X)$. Then, we partition all the data points to different layers according to their cost with respect to $\tilde{\theta}$. Specifically, we assign a point $x$ to the 0-th layer if $f(\tilde{\theta}, x) - \varrho < T$; otherwise, we assign it to the $\lfloor \log(\frac{f(\tilde{\theta}, x) - \varrho}{T}) \rfloor$-th layer. Let $L$ be the number of layers, and it is easy to see $L$ is at most $\log n + 1$. For any $0 \le j \le L$, we denote the set of points falling in the $j$-th layer as $X_j$. From each $X_j$, we take a small sample $C_j$ uniformly at random, where each point of $C_j$ is assigned the weight $|X_j|/|C_j|$. Finally, the union set $\bigcup_{j=0}^{L} C_j$ form our final coreset.

---

[3]In practice, we often add a positive penalty item to the objective function for regularization, so we can assume that $\inf_{\theta \in \mathbb{B}(\tilde{\theta}, \ell)} \frac{1}{n} f(\theta, X)$ is not too small.

**Theorem 6.** *Given a CnB learning instance $X$ with the objective function $f(\theta, X)$ as described in Definition 1, let* ddim *be the doubling dimension of the parameter space. The above coreset construction method GSP can achieve an $\varepsilon$-coreset of size $\Theta\left(\frac{\log n}{\varepsilon^2}\left(\mathtt{ddim} \cdot \log\frac{1}{\varepsilon} + \log\frac{1}{\eta}\right)\right)$ in linear time. The hidden constant of $|C|$ depends on the Lipschitz constant $\alpha$ and $\inf_{\theta \in \mathbb{B}(\tilde{\theta}, \ell)} \frac{1}{n} f(\theta, X)$.*

To prove Theorem 6, the key is show that each $C_j$ can well represent the layer $X_j$ with respect to any $\theta$ in the bounded region $\mathbb{B}(\tilde{\theta}, \ell)$. First, we use the continuity property to bound the difference between $f(\theta, x)$ and $f(\tilde{\theta}, x)$ for each $x \in X_j$ with a fixed $\theta$; then, together with the doubling dimension, we can generalize this bound to any $\theta$ in the bounded region.

## 5 Experiments

In this section, we illustrate the application of our proposed robust coreset method in machine learning. We leave the more detailed experimental results to the full version of this paper [44].

**Logistic regression (with outliers).** Given $x, \theta \in \mathbb{R}^d$ and $y \in \{\pm 1\}$, the loss function of Logistic regression is

$$f(\theta, x) = \ln(1 + \exp(-y \cdot \langle \theta, x \rangle)). \tag{22}$$

$k$**-median/means clustering (with outliers).** The goal is to find $k$ cluster centers $\mathtt{Cen} = \{c_1, c_2, \cdots, c_k\} \subset \mathbb{R}^d$; the cost function of $k$-median (*resp.* $k$-means) clustering for each $x \in \mathbb{R}^d$ is $f(\mathtt{Cen}, x) = \min_{c_i \in \mathtt{Cen}} d(c_i, x)$ (*resp.* $d(c_i, x)^2$), where $d(c_i, x)$ denotes the Euclidean distance between $c_i$ and $x$.

All the algorithms were implemented in Python on a PC with 2.3GHz Intel Core i7 CPU and 32GB of RAM. All the results were averaged across 5 trials.

**The algorithms.** We use the following three representative coreset construction methods as the black box in our hybrid framework for outliers. (1) UNIFORM: the simple uniform sampling method; (2) GSP: the generalized spatial partition method proposed in section 4.2; (3) QR: a QR-decomposition based importance sampling method proposed by [38] for logistic regression. For each coreset method name, we add a suffix "+" to denote the corresponding robust coreset enhanced by our hybrid framework proposed in section 3.

For many optimization with outliers problems, a commonly used strategy is alternating minimization (*e.g.,* [12]). In each iteration, it detects the $z$ outliers with the largest losses and run an existing algorithm (for ordinary logistic regression or $k$-means clustering) on the remaining $n - z$ points; then updates the $z$ outliers based on the obtained new solution. The algorithm repeats this strategy until the solution is stable. For logistic regression with outliers, we run the codes from the scikit-learn package[4] together with the alternating minimization. For $k$-means with outliers, we use the local search method [21] to seed initial centers and then run the $k$-means-- algorithm [12]. We apply these algorithms on our obtained coresets. To obtain the initial solution $\tilde{\theta}$, we just simply run the algorithm on a small sample (less than 1%) from the input data.

**Datasets.** We consider the following two real datasets in our experiments. The dataset **Covetype** [6] consists of 581012 instances with 54 cartographic features for predicting forest cover type. There are 7 cover types and we set the dominant one (49%) to be the positive samples and the others to be negative samples. We randomly take 10000 points as the test set and the remaining data points form the training set. The dataset **3Dspatial** [29] comprises 434874 instances with 4 features for the road information. To generate outliers for the unsupervised learning task $k$-means clustering, we randomly generate 10000 points in the space as the outliers, and add the gaussian noisy $\mathcal{N}(0, 200)$ to each dimension for these outliers. For the supervised learning task logistic regression, we add Gaussian noise to a set of randomly selected 10000 points (as the outliers) from the data and also randomly shuffle their labels.

**Results.** Table 1 and Table 2 illustrate the loss ratio (the obtained loss over the loss without using coreset) and speed-up ratio of different robust coreset methods. We can see that the robust coreset methods can achieve significant speed-up, and meanwhile the optimization qualities can be well preserved (their loss ratios are very close to 1). Figure 2(a) and 2(b) illustrate the performance

---

[4] https://scikit-learn.org/stable/

Table 1: Logistic regression on Covetype. $|C|$ denotes the coreset size.

| Method | $|C|$ | Loss ratio | Speed-up |
|---|---|---|---|
| GSP+ | $4 \times 10^3$ | 1.046 | $\times 26.9$ |
| GSP+ | $8 \times 10^3$ | 1.031 | $\times 19.19$ |
| UNIFORM+ | $4 \times 10^3$ | 1.134 | $\times 45.8$ |
| UNIFORM+ | $8 \times 10^3$ | 1.050 | $\times 29.1$ |
| QR+ | $4 \times 10^3$ | 1.025 | $\times 23.4$ |
| QR+ | $8 \times 10^3$ | 1.012 | $\times 17.9$ |

Table 2: $k$-means clustering on 3Dspatial with $k = 10$. $|C|$ denotes the coreset size.

| Method | $|C|$ | Loss ratio | Speed-up |
|---|---|---|---|
| GSP+ | $5 \times 10^3$ | 1.016 | $\times 41.1$ |
| GSP+ | $10^4$ | 1.008 | $\times 15.4$ |
| UNIFORM+ | $5 \times 10^3$ | 1.029 | $\times 78.9$ |
| UNIFORM+ | $10^4$ | 1.011 | $\times 46.9$ |

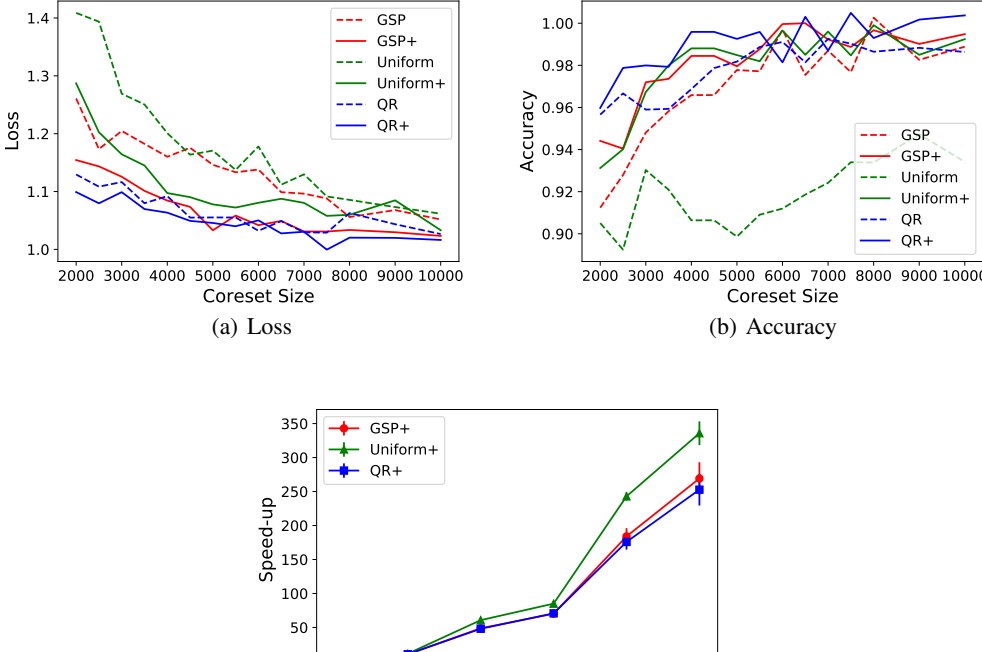

(a) Loss

(b) Accuracy

(c) Speed-up with the Merge-and-Reduce tree in the dynamic setting.

Figure 2: The performances of different coreset methods for logistic regression on Covetype. The results are normalized over the results obtained from the original data (without using coreset).

of the (robust) coreset methods with varying the coreset size. In general, our robust coreset can achieve better performance (in terms of the loss and accuracy) compared with its counterpart without considering outliers. Figure 2(c) illustrates the speed-up ratio of running time in the dynamic setting. Our robust coreset construction uses the merge-and-reduce tree method. When the update happens in one bucket, we perform a "bottom-up" re-construction for the coreset. We let the bucket size be $n/2^{h-1}$, where $h$ is the height of the tree; thus the higher the tree, the smaller the bucket size (and the speed-up is more significant). The results reveal that using the coreset yields considerable speed-up compared to re-running the algorithm on the entire updated dataset.

## 6  Conclusion

In this paper, we propose a novel robust coreset framework for the continuous-and-bounded learning problems (with outliers). Also, our framework can be efficiently implemented in the dynamic setting. In future, we can consider generalizing our proposed (dynamic) robust coreset method to other types of optimization problems (*e.g.,* privacy-preserving and fairness); it is also interesting to consider implementing our method for distributed computing or federated learning.

## Acknowledgment

We would like to thank the anonymous reviewers for their helpful suggestions and comments.

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
