# OpenReview forum: "Robust and Fully-Dynamic Coreset for Continuous-and-Bounded Learning (With Outliers) Problems"
_NeurIPS.cc/2021/Conference — NeurIPS 2021 Spotlight_

### Official Review · Reviewer_CN4f · 2021-07-10

**Rating:** 7
**Confidence:** 3

**Summary:**

The paper suggests a coreset construction framework for a family of functions that are Lipschitz continuous, smooth, and has Lipschitz continuous Hessian. The framework considers only continuous-and-bounded learning, i.e., optimization problems concerning bounded space of candidate solutions.

**Limitations And Societal Impact:**

One of the limitations of this work is that it handles bounded learning. It is interesting to show that under the umbrella of robust coresets, such limitation is somewhat necessary. Please provide a brief explanation of why it is not possible to remove such limitation, or where exactly in your method (or analysis), the necessity of this assumption is crucial?

In addition, Theorem 4 imposes an additional limitation. Specifically speaking, the total sensitivity that is being squared at the sample size. In recent works, e.g., "On coreset for Logistic Regression" and "Coresets for Near-Convex Functions", the total sensitivity was bounded from below $\Omega\left(\sqrt{n}\right)$ . This means the sample size in your case is $\Omega(n)$. I wonder if the total sensitivity that is obtained using your method is much smaller than the bounds achieved in the papers above due to the bounded learning assumption. Please briefly discuss this.

**Main Review:**

The paper suggests a robust coreset construction framework for several problems, including logistic regression and $k$-means clustering, under the umbrella of continuous-and-bounded learning. The method entails sensitivity sampling for "inliers" while using $\varepsilon$-sample for "outliers" to obtain a robust coreset. The ideas are interesting in nature and the work is not trivial.

I enjoyed reading the paper and the results are sound yet there are some misclarifications. Due to time constraints, I have managed to go over some of the proofs and they were sound other than the proof of Theorem 1 and Lemma 2; see my comment section below.

Comments:
- What is $Y$ at Line 168? It seems that $Y$ is the set $B$.
- There is an additional "s" at Line 300.
- It is very interesting that the sensitivity can be solved using quadratic fractional programming via a relaxation towards semi-definite programming. My question is, what is the total sensitivity in this case? what is the exponent in the $\mathrm{poly}(d)$ term at the time complexity?
- At Lemma 2, the bounds are not tight since the coreset is a robust coreset. This means that some of the transitions need to involve an additional multiplicative term of $O(1\pm\varepsilon)$. This is due to the fact that you end up dividing the sum of weights of instances of $C$ by the number of points in $X$. Through the literature of $\varepsilon$-coresets, and robust coresets as well, this term is in the range of $\left[ 1-\varepsilon, 1+\varepsilon\right]$.
- The proof of Theorem 1 is not clear as some of the transitions are not clear (see my first comment)
- Some of the coresets that you have mentioned handled instances of the problems that the paper discusses, e.g., logistic regression. However, I believe that the related work can be further enhanced, e.g., "Coresets for Near-Convex Functions" which proposes a unified framework for coreset construction for various loss functions including logistic regression, SVMs, and even robust $\ell_p$-regression. Using this paper as a black box to bound the sensitivity, how well your model will perform in terms of normalized loss error?
- Please move some of the graphs to the main manuscript to further enhance your results in the paper.




**Time Spent Reviewing:**

10

---

> ### Author Response · Authors · 2021-08-10
> **Rebuttal**
>
> We thank the review for the thoughtful comments. We summarize our responses in the following.
>
> > comments about notations, proofs and paper structure.
>
> We appreciate the reviewer's helpful comments, and we will improve our writing carefully.
>
> > comments about semi-definite programming
>
> The total sensitivity is the sum of the sensitivity of each point.
> Time complexity to compute this depends on the solving time of SDP, which is a bit subtle and we use $ploy(d)$ to roughly express the quantity. Fortunately, the semi-definite programming here is a convex one [2], so we can believe that this $ploy(d)$ is not very time-consuming.
>
> > Some of the coresets that you have mentioned handled instances of the problems that the paper discusses...
>
> The coreset for logistic regression in "Coreset for Near-Convex Functions" is a little different from ours and the coreset in [1]. Their method applies to logistic regression with a quadratic regularization term. However, the other two methods mentioned before is without regularization and thus can be also applied for any regularized version of logistic regression.
>
> > It is interesting to show that under the umbrella of robust coresets, such limitation is somewhat necessary...
>
> Even in some case without outlier, we cannot obtain a small coreset if we do not assume the boundness of the parameter. In the work of [1], they prove that there exists an instance of logistic regression in 2-dimensional
> Euclidean space such that any coreset for this instance consists of at least $\Omega(n/\log n)$ points.
>
> > if the total sensitivity that is obtained using your method is much smaller...
>
> The total sensitivity bound in Theorem 4 is data-dependent and there is no unified upper bound for general function $f$. As a complement, we introduce the GEL method in the later section.
>
>
>
> [1] On Coresets for Logistic Regression.
>
> [2] A Convex Optimization Approach for Minimizing the
> Ratio of Indefinite Quadratic Functions over an Ellipsoid.

---

> > ### Comment · Reviewer_CN4f · 2021-08-28
> > **Response**
> >
> > Thank you for the clarifications.
> >
> > In light of the author's clarifications and the points raised by other reviewers, I have decided to raise my score.

---

### Official Review · Reviewer_vMDk · 2021-07-11

**Rating:** 7
**Confidence:** 3

**Summary:**

This work provide coreset construction framework for functions that are Lipschitz continuous, smooth, and has Lipschitz continuous Hessian. The coreset in this paper is robust, i.e., it aims to handle outliers such that the coreset will be less biased than traditional coreset schemes.

**Limitations And Societal Impact:**

The authors adequately addressed the limitations, i.e., the coreset in this paper handle bounded learning (the candidate solution set for the problems discussed in the paper is bounded)

**Main Review:**

I enjoyed reading the paper, which suggests a robust coreset construction framework for multiple problems such like logistic regression and k-means clustering. The coreset uses sensitivity sampling for inliers, and $\delta$-sample for outliers. The ideas in this work are not trivial and the paper is well written, however I have some comments :
1) Line 177, what do you mean by "not integral"? did you mean integer?
2) I suggest moving the graphs to the paper. The graphs at supplementary material presents your efficiency against multiple candidates of coreset construction schemes.
3) I noticed some missing definitions, for example, in Theorem 1, line 168, what is Y? I did not completely understand the proof because of missing definitions in this Theorem.
4)Shouldn't the upper bound in Lemma 2 include $5 \epsilon$ instead of $3 \epsilon$ ?


**Time Spent Reviewing:**

6

---

> ### Author Response · Authors · 2021-08-10
> **Rebuttal**
>
> We thank the review for the thoughtful comments. We summarize our responses in the following.
>
> > comments 1,2,3,4
>
> We appreciate the reviewer's helpful comments, and we will improve our writing carefully by revising the notation, proofs and main paper.

---

### Official Review · Reviewer_uGsY · 2021-07-12

**Rating:** 6
**Confidence:** 4

**Summary:**

This paper presents an approach for constructing outlier-resilient coresets for continuous-and-bounded learning problems, which include logistic regression and k-means clustering. The authors build on recent work in dynamic coresets so that the constructions can be maintained in a fully-dynamic setting where input points can be inserted, deleted, and updated. The main idea of the outlier-resistant approach is to partition the original data set into two sets: one consisting of suspected outliers and the other for the “inliers.” For the outliers, the authors sample uniformly at random, and for the inliers, an existing coreset construction is used; these two subsamples are then merged to generate the outlier-resistant coreset. Under the assumptions of continuous and bounded learning, the authors also outline general QP-based and spatial-partition based methods to construct coresets.

**Limitations And Societal Impact:**

Please discuss limitations and potential negative societal impacts in Sec. 5 as reported in the Checklist.

**Main Review:**

Strengths

- The authors tackle an interesting and relevant problem of constructing coresets in the context of outliers. I also appreciate their effort in trying to make their approach as widely applicable as possible to common optimization problems and scenarios in large-scale machine learning.

- The technical results of the paper seem sound to the best of my knowledge.

- Although the approach of partitioning the input set and sampling from each set separately is not new in importance sampling (cf., stratification [2]), I found its application to outlier-resistant coresets to be interesting and novel.


Weaknesses

- The paper is quite dense and the organization and exposition could use improvements. Crucial details in certain parts of the paper (see comments regarding QP & SDP below) are not provided at all or deferred to the supplementary, but yet proofs of technical results (e.g., Lemma 2, Theorem 1, and Theorem 2) are provided inline -- taking up almost 2 pages total in the main paper. To me, this seems to distract from the flow of the paper, and also leads to important omissions. For example, the (only) empirical results of the work can only be found at the end of the supplementary material and additional details regarding the implementation of the method in practice are not provided (e.g., for a given sample size as in Sec. D of the supplementary, what $\varepsilon$ is used for the hybrid approach?). I would suggest deferring some of the technicalities to the supplementary in order to make room for further clarifying text around the main ideas and the experimental results (currently in Sec. D of the supplementary).

- Computing the sensitivities as described in Sec. 4.1 requires a reduction from a fractional Quadratic Program (QP) to a Semi-definite Program (SDP), with an overall running time of $O(n \text{poly}(d))$. Solving an SDP for every single point to compute the sensitivity upper bound seems to be highly costly in practice. Moreover, no details of how the sensitivity can be bounded via a QP formulation and SDP reduction are provided in the supplementary (despite the promise of additional details in Lines 307-310). Overall, I have difficulty seeing the added benefit of the sensitivity bounding approach presented in Sec. 4.1 relative to existing techniques for those specific problems (e.g., [3]). This also seems disjointed given the context of outlier-resistant coresets. Given the iterative SDP requirement, I am not sure how to make sense of the speed-up claims of up to $\approx 250$ in Fig. 4(c) (in the supplementary) relative to “training logistic regression on the original data” given that logistic regression training is quite efficient in practice and theory (e.g., via a gradient-based method).

- Relatedly, I am unclear about the connection to the GEL algorithm and why it is presented in conjunction with the QP-based method. Could we not bound the sum of sensitivities assuming that we have the optimizer for the sensitivity expression in Eq. 19 to obtain a data-independent upper bound on S? It seems to me that leaving the sum of sensitivities to be data-dependent and then theoretically bounding this quantity would yield better practical performance than an alternative approach that, in a sense, forces this sum to be data-independent. The sampling complexity of Theorems 4 and 5 also ignore factors $\alpha$ and $\inf_{\theta \in \mathcal{B}} \mathbb{f}(\theta, X)$, and it is not immediately clear to me when the bound would be vacuous: e.g., what happens if $\inf_{\theta \in \mathcal{B}} \mathbb{f}(\theta, X) \approx 0$?

- A more thorough discussion of the sampling complexity of Theorem 2 relative to that of related work in coresets with outliers would be illuminating (e.g., those of [1]).

- Lack of experimental details: the method requires knowing epsilon and $\tilde \theta$, among other hyperparameters that are required by the proofs and technical results. Yet, these values are not reported for the evaluations in the paper, and from the point of view of a practitioner, I am left confused as to what $\tilde \theta$ should be or how I would obtain it in practice. How were the results in Fig. 4 generated given a specified sample size?

- The experiments do not compare to prior work in coresets with outliers (e.g., [1]), among other related work that was cited in the introduction. Moreover, no computation time figures are reported for the offline setting, which makes it difficult to conclude that the proposed outlier-resilient approach leads to more accurate models (efficiently) in practice. Fig. 4(c) of the appendix reports the speedup in the dynamic setting as a function of the merge-and-reduce tree height, but only for the enhanced methods (with an “+” appended). It would have been illuminating to see computation time (x-axis) vs. normalized accuracy for the compared methods (similar to Fig. 4(b)). A discussion of why Uniform+ outperforms competing approaches in terms of the computational speedup (while performing on par with the coreset-based approaches) in Fig. 4 would be insightful.


Overall, I find the paper interesting and technically rigorous. However, I have qualms about the clarity and exposition of the paper and the practical merits of the proposed approach relative to prior work. At times, I felt like the paper tried to achieve too many things at once and some of the contributions seemed either disjointed or weren't well-motivated or emphasized (e.g., Sec. 4). I would suggest a clearer exposition in light of the comments above and more rigorous empirical evaluations that appear in the main paper.

[1] https://arxiv.org/pdf/1804.02530.pdf

[2] https://statweb.stanford.edu/~owen/mc/Ch-var-basic.pdf

[3] https://proceedings.neurips.cc/paper/2018/file/63bfd6e8f26d1d3537f4c5038264ef36-Paper.pdf


**Time Spent Reviewing:**

7

---

> ### Author Response · Authors · 2021-08-10
> **Rebuttal**
>
> We thank the reviewer for the thoughtful comments. We summarize our responses in the following.
>
> > The paper is quite dense and the organization and exposition could use improvements...
>
> We will improve our writing carefully and move some proofs to the supplement.
> > comments regarding QP and SDP
>
> The QFP (Quadratic Fractional Programming) based method is not compared in the experiment. It aims to provide a coreset method as general as possible. However, it is not suitable for speed-up, especially for problems that can be solved efficiently, because it has a higher computation time compared to other methods.
>
> > I have difficulty seeing the added benefit of the sensitivity bounding approach...This also seems disjointed given the context of outlier-resistant coresets.
>
> The QP-based method and GEL are both unified vanilla coreset methods for general problems. Together with those specific coreset methods, they are the black box in the construction of outlier-resistant coreset.
>
>
> > what happens if $\inf_{\theta} f(\theta, X)\approx 0 $?
>
> In the extreme case, $\inf_{\theta} f(\theta, X)$ could be close to zero. But in practice, for the real data, it is usually not the case. Furthermore, objective functions in machine learning often have a regularization term, which keeps itself from being close to 0.
> As a consequence, it is reasonable to assume that $\inf_{\theta} f(\theta, X)$ is larger than a threshold.
>
> > Lack of experimental details...
>
> The method to obtain $\tilde{\theta}$ is due to the specific problem. The method we use in the experiment is solving the problem on a tiny sample and set the result as $\tilde{\theta}$. Given a specific sample size, we construct different coreset with the same size. Then we compare the results (loss, accuracy on test set, and total running time) obtained from each coreset.
>
> > The experiments do not compare to prior work...
>
> The prior robust coreset method proposed in https://arxiv.org/pdf/1804.02530.pdf is exactly uniform sampling. The method proposed in https://proceedings.neurips.cc/paper/2018/hash/f7f580e11d00a75814d2ded41fe8e8fe-Abstract.html cannot control the size of coreset directly thus is not suitable for comparison.
>
> The speed-up has included the coreset construction time and we can find that the robust coreset is efficient in practice. Uniform+ method usually has lower  construction time (uniform sampling is easy to implement and has small time complexity). When all coresets have the same size, it is not surprising that uniform+ is the fastest one.

---

> > ### Comment · Reviewer_uGsY · 2021-08-26
> > **Thanks for the clarifications**
> >
> > Thank you for your detailed responses. I appreciate the clarifications.
> >
> > In light of the author's clarifications and the points raised by other reviewers, I have decided to raise my score. This is mainly due to the contextualization of the contributions relative to those of prior work. I still have qualms about the empirical results of this paper and echo the comments of Reviewer Pcf3 regarding the lack of compelling experimental motivation and results.

---

> > > ### Author Response · Authors · 2021-08-28
> > > **Response to comment about experiment**
> > >
> > > We thank the reviewer for the feedback. We apologize that the experimental results are placed in the supplement due to the space limit. We will modify our paper's organization and add them back. We will also compare our method with some papers mentioned by Reviewer Pcf3   (https://arxiv.org/abs/2012.10630, https://arxiv.org/pdf/2011.07451.pdf, https://arxiv.org/pdf/2103.00123.pdf). Some of them are published very recently, but we will conduct more comparisons in our paper.

---

### Official Review · Reviewer_4yNi · 2021-07-12

**Rating:** 6
**Confidence:** 5

**Summary:**

This paper studies coresets for continuous-and-bounded learning problems. In the optimization version of the problem, given a weighted input data set X with weight w : X \to \mathbb{R}_+, the goal is to find a parameter \theta in some parameter space P such that \sum_{x \in X} w(x) \cdot f(\theta, x) is minimized where f : P \times X \to \mathbb{R}_+ is a cost function. Furthermore, the continuous-and-boundedness condition requires that the parameter space P form a metric space (so there is an implicit associated distance function on the parameters), and that a) there exist l, \alpha > 0, such that all parameters in P are contained in a ball of radius l, and b) f(\cdot, x) is \alpha-Lipschitz.

An \epsilon-coreset for this problem is defined as a weighted subset S \subseteq X of the data set with weight u : S \to \mathbb{R}_+, such that for every parameter \theta \in P, the total cost on S is close to that on X, i.e., \sum_{x \in S} u(x) \cdot f(\theta, x) \in (1 \pm \epsilon) \sum_{x \in X} w(x) \cdot f(\theta, x).

The paper also studies coresets for the outlier version of the problem, where we are additionally given a parameter z (the number of outliers) and the cost becomes f_z(\theta, X) = \min_{O \subset X, |O| = z} f(\theta, X \setminus O). Coresets are defined similarly for the outlier version, except that this paper allows bi-criteria guarantee: a (\beta, \epsilon)-coreset preserves the cost for every \theta, such that the cost is within [f_{(1+\beta)z}(\theta, X), f_{(1-\beta)z}(\theta,X)] (the exact definition is subject to proper normalization).

Several results are presented. For the outlier version, the first result shows that uniform sampling yields a (\beta, 0)-coreset of size \tilde{O}(n / (\beta z))^2, provided that the VC-dimension for the (metric ball) range space induced by {f(\cdot, x) : x} is bounded. This result is weak and is inadequate for small z (even for moderately small z = O(\sqrt{n})). The second result is more interesting and it gives a reduction that reduces the outlier version to the non-outlier version. Intuitively, it “glues” a black-box coreset for the non-outlier version and a uniform sampling based coreset. Specifically, it yields a (\beta, \epsilon) coreset of size T + min{ O((\beta\epsilon)^{-2} vcdim), O(z / epsilon)}, where T is the size of some \epsilon-coreset in the non-outlier setting, and vcdim is the VC-dimension for the range space induced by {f(\cdot, x) : x}. The paper also gives a fully-dynamic algorithm that maintains the “glued” coreset.

Besides, two coreset results for the non-outlier version are presented. The first one is based on the well-known importance sampling framework [11], but instead of analyzing the VC-dimension of the range space, this paper shows that the VC-dimension term could be replaced by the doubling dimension of the parameter space P. However, the authors were not able to give an upper bound for the total sensitivity, and therefore this first result does not yield an explicit coreset size bound. Then the other result is based on a spatial partition method proposed in [9], which helps to gets rid of the total sensitivity bound, but it introduces a log n factor.


**Limitations And Societal Impact:**

Yes.

**Main Review:**

Originality & significance:

The main novelty is the initiative of a unified framework for coresets for continuous-and-bounded learning with outliers. The main result which reduces the outlier version to the non-outlier version is novel, and I do think this reduction seems to be strong enough to imply interesting new bounds for concrete instances of continuous-and-bounded learning problems. Unfortunately, authors did not give any such examples, which makes me uncertain about the usefulness of this new unified theory. In particular, the theorem requires the VC-dimension of the f-induced range space to be bounded – can you show an interesting example of continuous-and-bounded learning problem that has VC-dimension bounded? It also makes it hard to tell whether this result can improve previous works without such examples.

An additional comment is that, your coreset is bi-criteria for the outlier problems – is the violation in the number of outliers necessary? Can you possibly justify this?

As another contribution, several coreset constructions for general continuous-and-bounded learning (without outliers) are given, and all of the proposed coresets depend on the doubling dimension of the parameter space. But I’m not sure if considering doubling dimension on the parameter space makes sense – indeed, doubling dimension is an intrinsic dimensionality measure, but it is usually applied on data set instead of parameter space. In fact, parameter space is data-independent and usually depends on the problem itself, which does not seem to benefit from low intrinsic dimension.

Technically, many results are obtained by combining existing techniques in a straightforward way, and I don’t see much novelty. For instance, the dynamic algorithm follows almost immediately from previous works, and it doesn’t add much to the paper; also, the two coresets for non-outlier version of the problem are very much standard and follows from previous framework with only minor modifications.


Quality:

I think the paper is technically sound, and all major claims are reasonably supported by proofs. However, a key missing part of the paper is some concrete applications of the unified theory – maybe some new bounds for specific learning problems? Or experiments that validate the theory in a certain way? (I do see some experiment results were reported in the supplementary, but unfortunately the findings of the experiments are not discussed in the main text at all.) This would make the theory more useful to the machine learning community.

Clarity:

The paper is overall very well written. It does a good job to provide necessary information for (at least expert) readers to understand why each claim holds, without needing to read proofs.

Other comments:

1.	Line 31 “noise” -> “noisy”
2.	Line 61 should be “continuous-and-bounded”
3.	In Definition 2, why use a different font for f? In general, please make sure you use the font consistently (which seems not the case in the paper).
4.	In Definition 4, why do we need the assumption that \mathcal{X} is a metric space? I don’t think you need this.
5.	Line 164, what do you mean by “\delta-approximate subset”? I didn’t find the definition.
6.	Line 165, the use of O in “O = ab” seems not a good idea, since O is usually used as in the big O notation.
7.	Line 168, what is Y?
8.	Theorem 2, line 223, what’s the third parameter 2\xi(l)? I think the notation only has two parameters?
9.	Line 228, can you explain why f_z(\theta, C) = f(\theta, C_I + C_{II})? In particular, why don’t we need C_{III} and C_{IV}?
10.	In the statement of Theorem 3, the VC dimension should be a uniform upper bound for weighted range spaces over all weights.
11.	In the statement of Theorem 4, line 321, you mentioned the hidden constant depends on \inf_{\theta} f(\theta, X) – why this is bounded?


**Time Spent Reviewing:**

5

---

> ### Author Response · Authors · 2021-08-10
> **Rebuttal**
>
> We thank the reviewer for the thoughtful comments. We summarize our responses in the following.
>
> > I do think this reduction seems to be strong enough to imply interesting new bounds...
>
> > a key missing part of the paper is some concrete applications of the unified theory – maybe some new bounds for...
>
> Previous researches on robust coreset are substantially limited compared to the non-outlier version and concentrate on clustering-like problems. As far as we know, we are the first one to give a robust (outlier-resistant) coreset for a broader range of problems including truth discovery, logistic regression, and general Bregman clustering which are discussed in section A of the supplement. The bound can be computed with the bound of corresponding "dimension"s discussed below.
>
> > can you show an interesting example of continuous-and-bounded learning problem that has VC-dimension bounded?
>
> The VC-dimension of the $f$-induced range space has been discussed in many previous works. In Lemma 10, Lemma 11 from [1], they show that as for logistic regression, the VC-dimension of the $f$-induced range space is $d+1$. It is not hard to prove that for truth discovery, it is $ O(d\log d) $ and $O(kd\log d)$ for $k$-median/means.
>
> > is the violation in the number of outliers necessary?
>
> The violation in the number of outliers is not necessary whereas the price is a bigger coreset size. Our $(\beta,\varepsilon)$-coreset method does not always have this violation since $\beta$ can be 0 as mentioned in Theorem 2, where the additional size is $O(z/\varepsilon)$. If no violation of the number of outliers is permitted, what is the lower bound of the robust coreset size? We believe it is an interesting question to study in the future.
>
> > if considering doubling dimension on the parameter space makes sense...
>
> We use doubling dimension to make a unified statement. If the parameter space is Euclidean space $\mathbb{R}^d$, its doubling dimension is $\Theta(d)$. For the problems like clustering in the metric space, the parameter (solution) space is also the data space, which may have a low intrinsic dimensional structure.
>
> > many results are obtained by combining existing techniques in a straightforward way...
>
> The merge-and-reduce framework (used in the dynamic setting) for the robust coreset with outliers is not immediate from the non-outlier version. One of the most important properties of vanilla coreset is that it can be distributedly constructed, specifically, by the merge-and-reduce framework. Because of this, a coreset method can induce a streaming algorithm [2] and a dynamic algorithm [3] of the original problem. Nevertheless, this cannot be done for the general robust coreset since we do not know the number of outliers in each distributed part. Our hybrid framework firstly overcomes this thus gives a robust streaming algorithm and a robust dynamic algorithm for the original optimization with outliers. Moreover, we are the first (to our best knowledge) to consider and propose the dynamic algorithm enabling the variety of $z$.
>
> The following are short responses to comments:
>
> - comment 1,2,3,5,6,7: we use another $f$ to denote the average form as mentioned in Table 1. We appreciate the reviewer's helpful comments, and we will improve our writing carefully.
>
> - comment 8:  we use $\xi(\ell)$ to state the result of different Lipschitz continuous functions (defined in Line 92~96) uniformly. As mentioned in line 219, the explicit form of $\xi(\ell)$ depends on which kind of Lipschitz continuous function it is.
>
> - comment 9: $f_z(\theta, C) = f(\theta, C_I + C_{II})$ comes from the definition of $C_I,C_{II},C_{III}$ and $C_{IV}$. Because $C_{III}+C_{IV}$ is the real $z$ outliers w.r.t $\theta$.
>
> - comment 10: We omit the statement of the weighted case for simplicity and it is not hard to extend this as mentioned in line 111.
>
> - comment 11: In the extreme case, $\inf_{\theta} f(\theta, X)$ could be close to zero. But in practice, for the real data, it is usually not the case. Furthermore, objective functions in machine learning often have a regularization term, which keeps itself from being close to 0.
> As a consequence, it is reasonable to assume that $\inf_{\theta} f(\theta, X)$ is larger than a threshold.
>
>
> [1] On Coresets for Logistic Regression https://proceedings.neurips.cc/paper/2018/file/63bfd6e8f26d1d3537f4c5038264ef36-Paper.pdf
>
> [2] On coresets for k-means and k-median clustering https://dl.acm.org/doi/10.1145/1007352.1007400
>
> [3] Fully-Dynamic Coresets https://drops.dagstuhl.de/opus/volltexte/2020/12923/

---

> > ### Comment · Reviewer_4yNi · 2021-08-25
> > **Increasing my score**
> >
> > Thanks for the detailed response.
> >
> > I'm convinced that the unified theory has many applications and that there are indeed technical challenges in applying merge-and-reduce in the outlier setting.
> >
> > However, I still think considering doubling dimension in the parameter space doesn't make much sense. It's correct that for clustering problems the doubling dimension of the parameter space is O(d) if the input is in R^d, but what if the input dimension d is huge, while the data has a low doubling dimension? (Note that this "low doubling dimension in high dimensional data" is exactly where considering doubling dimension makes sense - it measures the low "intrinsic" dimension of the data.) I think the doubling dimension of the parameter space should still be d instead of the small doubling dimension of the data.
> >
> > Overall, I'm willing to increase my score, but I'm still concerned with certain issues in the paper which makes me hesitant to give a clear recommendation for acceptance.

---

> > > ### Author Response · Authors · 2021-08-28
> > > **Response to comment about doubling dimension**
> > >
> > > We thank the reviewer for the feedback. To see why we consider the doubling dimension in the parameter space, we can take  "sparse optimization" as another example. In many machine learning tasks, we often restrict our parameter vector $\theta$ to be sparse.
> > >
> > > Let the parameter space be $ \mathbb{R}^D $, and we restrict $\theta$ to be $k$-sparse (i.e., at most $k\ll D$ non-zero entries). It is easy to see the domain $\mathcal{A}$ of $\theta$ is a union of ${D\choose k}$ $k$-dimensional subspaces, and thus the doubling dimension of $\mathcal{A}$ is $O(k\log D)$ which is much smaller than $D$ (each ball of radius $r$ in a $k$-dimensional subspace can be covered by $2^{O(k)}$ balls of radius $r/2$; therefore, each ball of radius $r$ in $\mathcal{A}$ can be covered by ${D\choose k}\cdot 2^{O(k)}\leq D^k\cdot 2^{O(k)} = 2^{O(k\log D)} $ balls of radius $r/2$; overall, the doubling dimension of $\mathcal{A}$ should be $O(k\log D)$).

---

### Official Review · Reviewer_Pcf3 · 2021-07-20

**Rating:** 6
**Confidence:** 4

**Summary:**

The goal of this paper is to investigate a coreset approach for robust supervised learning specifically with outliers. The authors study this for logistic regression and k-means clustering. This paper is mainly theoretical and the empirical results are mainly in the appendix.

**Ethical Concerns:**

This paper is mainly theoretical.

**Limitations And Societal Impact:**

This paper is mainly theoretical.

**Main Review:**

Pros:
a)	Paper provides an extensive theoretical study analyzing the robust coreset selection framework and proved a theorem stating that the proposed framework can successfully compute a (beta, epsilon) coreset.
b)	Provided experimental results show that the proposed coreset selection framework achieves better speedup and accuracy computer to other coreset selection methods.

Cons:
a)	The theoretical assumption of boundedness is too restrictive for deep networks. Hence, the proposed framework may not work well in a deep learning scenario. It would be nice to have some results using deep models even if the proposed method does not work so that others have an understanding of the limitations of the proposed approach.

b)	In the experimental section (in the appendix), the authors mentioned speedups and accuracies achieved. It would be interesting to know how many actual outliers the proposed framework was able to select or filter during training.

c)	The used coreset selection methods for experiments are simple. Hence, it would be interesting to know how the proposed coreset selection framework compare to other robust coreset selection frameworks like GLISTER (https://arxiv.org/abs/2012.10630), Coresets for Robust Training of Neural Networks against Noisy Labels (https://arxiv.org/pdf/2011.07451.pdf) and Grad-Match (https://arxiv.org/pdf/2103.00123.pdf)  in terms of speedups and accuracies. It would also be interesting to find if the proposed approach can also be applied with the above-mentioned robust coreset selection methods.

d)  The major limitation of this work is the lack of solid experiments. I would have expected more empirical results particularly given the nature of the problem and not have the experiments relegated to the Appendix. I would encourage the authors to discuss how their approach compares to other subset selection and coreset approaches for robust learning (like those mentioned above).

Post Rebuttal Update:
The authors have clarified a few of the issues pointed out. However, a major criticism of this work is the lack of experiments. This, in my opinion, can be a limiting factor for the impact of this work, mainly because a lot of work in this area is empirical and an understanding of how an approach like this performs empirically is critical in understanding its place among the state-of-the-art. However, I do think this paper contributes from a theoretical perspective, and I'll keep my score of weak acceptance. I would strongly encourage the authors to add empirical evaluation in the main text if given a chance.

**Time Spent Reviewing:**

3-4 hours

---

> ### Author Response · Authors · 2021-08-10
> **Rebuttal**
>
> We are very thankful to the reviewer for the helpful suggestions. We summarize our responses in the following.
>
> > It would be interesting to know how many actual outliers the proposed framework was able to select or filter during training.
>
> The number of actual outliers that this framework can select depends on the distribution of outliers.
> Specifically, if the outliers are generated from some probabilistic distribution $ D $, we found that the number of actual outliers selected by this framework arises with the variance of the distribution $ \sigma^2(D) $. And this conclusion is consistent with our intuition. We did not officially outline this conclusion in this version but we will modify our paper with a more detailed experiment part.
>
> >"......Hence, the proposed framework may not work well in a deep learning scenario....."
>
> Thanks for this question, and we believe it is very interesting to study coresets for deep learning in future. There are still several challenging obstacles, e.g., the VC dimension for deep learning objective may be still unclear. Also, since deep learning usually involves a large number of parameters, it may be more important to consider compression (like network pruning) of network structure.

---

> > ### Comment · Reviewer_Pcf3 · 2021-08-25
> > **Thanks for your response**
> >
> > Thanks for your response. I think this paper does make a contribution theoretically. However, the key weakness of this work in my mind is that wearing the hat of a practitioner, I do not know how well their approach works in real-world problems nor is it clear how this performs compared to other state-of-the-art approaches for robust learning. Given this, I'll keep my score of weak accept, but I would strongly encourage the authors to consider strengthening the empirical component of their work.

---

> > > ### Author Response · Authors · 2021-08-28
> > > **Response to comment about experiment**
> > >
> > > We thank the reviewer for the feedback. We apologize that the experimental results are placed in the supplement due to the space limit. We will modify our paper's organization and add them back. We also thank the reviewer for mentioning the papers   (https://arxiv.org/abs/2012.10630, https://arxiv.org/pdf/2011.07451.pdf, https://arxiv.org/pdf/2103.00123.pdf). Some of them are published very recently, but we will conduct more comparisons in our paper.

---

### Decision · Program_Chairs · 2021-09-27

**Decision:**

Accept (Spotlight)

**Comment:**

This is a beautiful paper about coresets for handling outliers which will probably inspire many future related papers.
While there are concerns regarding experimental results (that are hidden in the supp. material), the theoretical contribution with an algorithm that it is not hard to implement is strong enough for such a fundamental problem in machine learning.
Please move some more experiments to the main paper in order to attract the practitioners.

Also please add the following references and maybe some more:
https://www.mdpi.com/1999-4893/13/12/311
https://dl.acm.org/doi/10.5555/1347082.1347173